

# Velocity increases at Cook Glacier, East Antarctica linked to ice shelf loss and a subglacial flood event

Bertie W.J. Miles[1*], Chris R. Stokes[1], Stewart S.R. Jamieson[1]

[1]*Department of Geography, Durham University, Science Site, South Road, Durham, DH1 3LE*

*Correspondence to: a.w.j.miles@durham.ac.uk*

**Abstract:** Cook Glacier drains a large proportion of the Wilkes Subglacial Basin in East Antarctica, a region thought to be vulnerable to marine ice sheet instability and with potential to make a significant contribution to sea-level. Despite its importance, there have been very

few observations of its longer-term behaviour (e.g. of velocity or changes at its ice front). Here we use a variety of satellite imagery to produce a time-series of ice-front position change from 1947-2017 and ice velocity from 1973-2017. Cook Glacier has two distinct outlets (termed East and West) and we observe the near-complete loss of the Cook West Ice Shelf at some time between 1973 and 1989. This was associated with a doubling of the velocity of Cook West

glacier, which may also be linked to previously published reports of inland thinning. The loss of the Cook West Ice Shelf is surprising given that the present-day ocean-climate conditions in the region are not typically associated with catastrophic ice shelf loss. However, we speculate that a more intense ocean-climate forcing in the mid-20th century may have been important in forcing its collapse. Since the loss of the Cook West Ice Shelf, the presence of landfast sea-ice

and mélange in the newly formed embayment appears to be important in stabilising the glacier front and enabling periodic advances. We also observe a short-lived increase in velocity of Cook East between 2006 and 2007 which we link to the drainage of subglacial Lake Cook. Taken together, these observations suggest that the velocity, and hence discharge, of Cook Glacier is highly sensitive to changes at its terminus but a more detailed process-based analysis

of this potentially vulnerable region requires further oceanic and bathymetric data.

## 1. Introduction

Ice which is grounded well below sea level in the marine basins of Antarctica is potentially vulnerable to marine ice sheet instability. This is because an initial grounding line retreat into



deeper water can create an unstable and self-sustaining feedback leading to increased ice discharge, inland thinning, and a rapid sea level contribution (Hughes, 1981; Schoof, 2007). Floating ice shelves are crucial to the stability of ice streams and outlet glaciers that drain marine basins because they can exert an important buttressing effect (Furst et al., 2016). Thinning or retreat of these ice shelves reduces their ability to restrain flow from the ice sheet

(Pritchard et al., 2012). This is evident in parts of the West Antarctic Ice Sheet (WAIS) where the feedbacks resulting from the rapid thinning of ice shelves (Paolo et al., 2015) has resulted in an increased discharge of ice into the ocean (Mouginot et al., 2014). This oceanic-driven thinning of ice shelves may have destabilized the Thwaites Glacier basin, where marine ice sheet instability may already be underway, and which might undermine much of the WAIS

over the coming decades to centuries (Joughin et al., 2014).

The Wilkes Subglacial Basin (WSB) in East Antarctica contains 3-4 m of sea level equivalent of ice grounded below sea level (Mengel and Levermann, 2014). Geological evidence suggests the WSB may have made substantial sea level contributions during the warm interglacials of the Pliocene (Williams et al., 2010; Cook et al., 2013), which are thought to represent the best

analogue for near-future climates under continued global warming. Indeed, numerical ice sheet models predict future sea-level contributions from the WSB, but the magnitude and timing of the contributions varies (Golledge et al., 2015; Ritz et al., 2015; DeConto and Pollard, 2016). Furthermore, dynamical modelling of the present day ice-sheet margin of the WSB shows that its stability might be controlled by a relatively small band of coastal ice (~80 mm SLE), which

is preventing a self-sustained discharge of the entire basin (Mengel & Levermann, 2014). The majority of this coastal band of ice is drained by Cook Glacier (Mengel & Levermann, 2014), which is one of the largest in Antarctica. Its current configuration consists of two distinct distributaries: Cook East and Cook West (Fig. 1). Cook East flows into a large 80 km long ice shelf, whereas Cook West terminates close to, or at, its grounding line (Fig. 1). Despite having

one of the largest annual discharges of any Antarctic outlet glacier (Rignot et al., 2013) (~36 Gt a$^{-1}$), and given its potential significance to the stability of the WSB, there have been very few observations of its recent behaviour. Along with Totten Glacier, it was specifically highlighted in the most recent IPCC report (2013) as being potentially vulnerable to marine ice sheet instability but, unlike Totten, there has been no obvious changes in ice shelf thickness or

ice surface elevation over the past decade (Pritchard et al., 2009; McMillan et al., 2014; Paolo et al., 2015). However, some studies have previously highlighted Cook as a region of modest inland thinning (e.g. Shepherd and Wingham, 2007) and Frezzotti et al. (1998) reported a major



retreat of the Cook West Glacier, which others have suggested might be flowing too fast to be in balance (Rignot, 2006). In this paper, we report on the long-term changes in Cook Glacier

by combining measurements of ice-front position from 1947-2017, together with glacier velocity estimates from 1973-2017 from optical based feature-tracking. Our results indicate that despite little change over the past decade, there has been a long-term increase in the velocity of both Cook East and Cook West Glaciers that can be linked to changes at its ice-front.


## 2. Methods

### 2.1 Ice-front position change

We revisit and extend the results of Frezzotti et al. (1998) by using a combination of oblique aerial photography from 'Operation Highjump' in 1947, ARGON, RADARSAT, ASTER,

Landsat and WorldView-2 satellite imagery to create a 70 year time-series of ice-front positon change from 1947-2017 for both Cook West and Cook East glaciers (Table S1). Changes in ice-front position were quantified by the well-established box method which takes into account uneven changes along the ice-front (e.g. Moon and Joughin, 2008). Errors using this method arise from the co-registration of satellite images and the manual digitization of the ice-front,

which we estimate to be a combined 1.5 pixels (22.5 – 90 m). These errors are in the range of similar studies and are insignificant when quantifying ice-front position change of large Antarctic outlet glaciers (Miles et al., 2013; 2016). Because the 1947 aerial photographs were taken at an oblique angle we estimate the ice-front positon based on the ice-front positon relative to stable features which have not moved over time (e.g. ice rises). This creates larger

uncertainties compared with measurements from orthorectified satellite imagery. We estimate these uncertainties at ~±2 km.

### 2.2 Glacier velocity from feature tracking

Estimates of glacier velocity were derived using COSI-Corr (Co-registration of Optically Sensed Images and Correlation) feature-tracking software (Leprince et al., 2007; Scherler et

al., 2008). This software tracks spectral signatures which relate to features on the glacier surface that can be identified in multiple images through time, and it has been shown to be one of the most robust methods of glacier velocity mapping (Heid and Kaab, 2012). It requires pairs





of co-registered optical cloud-free images which are spaced close enough in time for surface features to be identified in both images. In this study, the temporal resolution of image pairs

was largely determined by the availability of appropriate satellite imagery, which was generally sparse due to a combination of poor coverage and persistent cloud cover. However, by using a combination of Landsat-1, Landsat-4, Landsat-7, ASTER and Landsat-8 we were able to create a velocity time series from 1989-2016 for Cook East (Table S2) and 1973-2017 for Cook West (Table S3). Image pairs were typically spaced 1 year ± 100 days apart, which is a suitable gap

for the preservation of surface features. The exception to this was in 1973-74 where image availability only allowed temporal gap of 73 days (Table S3).

The COSI-Corr procedure first requires the accurate co-registration of image pairs. Co-registration was achieved by manually co-registering image pairs using a combination of nunataks and the boundaries of ice rises, which are known to be stable features over time.

Because these features were relatively common in the vicinity of Cook Glacier, images pairs were able to be co-registered to an estimated accuracy of 1 pixel. We used a window size of 256 x 256 pixels and a grid size of 20 x 20 pixels to detect surface displacement and produce velocity grids. Error in surface displacement was estimated at 0.5 pixels, which is consistent with other studies using this method (Scherler et al., 2008; Heid and Kaab, 2012). Total error

ranged from ±51 m yr$^{-1}$ in 1989 to between ±19 and ±24 m yr$^{-1}$ from 2000-2017 (Table S2 and S3). The coarser resolution and closer temporal resolution of the 1973/4 Landsat-1 image pairs resulted in a considerably higher error of ±450 m yr$^{-1}$ (Table S3).

Post-processing of ice velocity grids can reduce noise and remove erroneous pixels (e.g. Mouginot et al., 2017). We removed pixels which were greater than ±25% of the MEASURES

ice velocity product (Rignot et al., 2011) in velocity grids from 2000-2017 and ±40% in the velocity grid from 1989 to account for any larger changes in glacier velocity. For the 1973/4 velocity grid, we filtered out all pixel values below 450 m yr$^{-1}$ to account for the larger error. We then applied a low pass filter to all velocity grids to create the final products (Fig. 2 To create the velocity time series we extracted the mean value of pixels within a defined box across

all epochs, in each epoch all pixels were sampled within the defined box i.e. there were no rejected pixels. For Cook East the defined box was on a section across the grounding line (Fig. 2), because Cook West terminates close to its grounding line we extracted velocities 2 km upstream (Fig. 2).

### 2.3 Ice-front advance rate



Preliminary inspection of the imagery clearly indicated that there have been no major calving
events on the Cook East Glacier since 1973 because the shape of the ice margin is unchanged.
Thus, we were able to create a time series of the rate of ice-front advance between 1973 and
2016. Although it is not a direct measurement of glacier velocity, the rate of ice-front advance
is helpful in enabling additional independent estimates of ice advance (a proxy for ice velocity
at the terminus if no major calving events have taken place) further back in time (i.e. between
1973 and 1989) and allows additional measurements to be made in the 1990s (Table S4). Ice-
front advance rate was quantified by dividing ice-front position change by the number of days
between image pairs. Taking into account the error of 1.5 pixels associated with co-registration
and manual mapping, errors were estimated between $\pm 1$ to $\pm 86$ m yr$^{-1}$ with range in error
accounting for the varying spatial resolution of images and the temporal gap between image
pairs (Table S4).

## 3. Results

### 3.1 Cook East

The Cook East Ice Shelf last underwent a major calving event at some point between 1963 and
1973 (Fig 3 & 4). This calving event resulted in the retreat of its ice-front deep into the
constrained section of its embayment (Fig. 3). Since 1973 it has advanced ~31 km and there
have been no major calving events. Through extrapolating the rate of advance between 1947
and 1963 we estimate that its maximum ice-front extent prior to its calving at some point
between 1963 and 1973 was similar to its 2015 position (Fig. 4). This means that its present-
day ice-front is likely further advanced than the point at which it last underwent a major calving
event/retreat.

The velocity of Cook East increased by 19% from 416 $\pm 51$ m yr$^{-1}$ in 1989 to 496 $\pm 19$ m yr$^{-1}$ in
2000/01 (Fig 5). Throughout 2001 to 2016 velocity remained consistent with an average speed
of 489 m yr$^{-1}$, with little year to year deviation. The only exception to this was between 2006
and 2007 where Cook East was flowing 12% (545 $\pm 22$ m yr$^{-1}$) faster than its 2001-2016
average. Velocity profiles across the Cook East Ice Shelf show similar patterns (Fig. 6), with
the exception of 1989 and 2006-07, all profiles are clustered in a narrow band. In 1989 velocity
was anomalously slow across the entire ice shelf and between 2006 and 2007 velocity was



anomalously fast. Notably, these patterns also persist several kilometres upstream of the grounding line (Fig. 6).

There was little change in the rate of mean ice-front advance between 1973 and 1997. However, from 1997-2000 (720 ±20 m yr$^{-1}$) and 2002-2006 (749 ±8 m yr$^{-1}$) there was a consistent increase in the rate of ice-front advance (Fig. 5). This is consistent with velocity estimates from the

grounding line which show an increase in velocity between 1989 and 2001. Throughout 2002-2016 there were small internannual variations in ice-front advance rate, with no obvious trend. In a similar manner to velocity estimates from the grounding line, the only exception to this was between 2006 and 2007 where the ice front advanced at 792 ±30 m yr$^{-1}$, higher than the 2002-2016 average (752 m yr$^{-1}$) (Fig. 5).

**3.2 Cook West**

From 1947 to 2018 the Cook West ice-front retreated approximately 34 km (Fig. 7 & 8). This retreat largely occurred in two stages, with retreat initiating between 1947 and 1963 when Cook West retreated 20 km, before stabilizing between 1963 and 1973 when the ice-front retreated 2.8 km. From 1973 to 1989 the remaining section of Cook West's Ice Shelf retreated 13 km

close to, or onto, its present grounding line (Fig. 7 & 8). This resulted in an estimated total loss of 1,200 km$^2$ of ice shelf between 1947 and 1989. The large retreat of 5 km between November 1973 and January 1974 (Fig. 7 & 8) suggests that this retreat was more likely to have occurred in the mid-1970s. Since 1989, observations show relatively little change, with only minor fluctuations (~3 km) in ice-front position. Perhaps surprisingly, we observe no signs of a re-

advance of Cook West Glacier comparable to its pre-1989 ice-front position.

The velocity of Cook West Glacier increased from 692 ±450 m yr$^{-1}$ in 1973/4 to 1438 ±51 m yr$^{-1}$ in 1989 (Fig. 8). Although the error associated with the 1973/74 measurement is high (±450 m yr$^{-1}$), it demonstrates a clear increase in velocity, which coincides with the retreat of the Cook West ice-front (Fig. 8). There were small variations in the velocity of Cook West between

2001 and 2017, with no velocity estimates deviating from ±5% of the 2001-2017 mean (1368 m yr$^{-1}$). Between 2001 and 2017, Cook West was flowing fastest from 2001-2002 at 1463 ±24 m yr$^{-1}$ and slowest from 2016-2017 at 1306 ±22 m yr$^{-1}$ (Fig. 8).

**4. Discussion**

**4.1 Cook East**



### 4.1.1 Long-term behaviour the Cook East Ice Shelf

The calving of the Cook East Ice Shelf between 1963 and 1973 resulted in the retreat of its ice-front deep into the constrained section of its embayment (Fig. 3). This is unusual in the context of large Antarctic ice shelves where calving events typically occur within the bounds of the

unconstrained section of ice shelves (Miles et al., 2013). The return period of any potential calving cycle at Cook East may be too long to determine if this relatively deep retreat into the embayment is typical of its normal behaviour (Fig. 4). Based on the morphology and size of an iceberg located near the Mertz Ice Tongue in satellite imagery in 1984, Frezzotti et al. (1998) estimated that the calving of Cook East between 1963 and 1973 occurred in the early 1970s.

This would imply that its current ice-front position is further advanced than its last calving event (Fig. 4). However, an inspection of the current morphology of the Cook East Ice Shelf reveals no obvious signs of an imminent calving event and we suggest another calving event is at least several years away.

We observe an increase in the velocity of the Cook East Ice Shelf from 416 ±51 m yr$^{-1}$ in 1989

to 496 ±19 m yr$^{-1}$ in 2000-2001 (Fig 5). The rate of the Cook East ice-front advance also increases between 1989 and 2002 but, notably, most of this acceleration is concentrated between 1997 and 2002 (Fig. 5). The rate of ice-front advance is not a direct estimate of velocity because there are processes such as longitudinal stretching which can result in changes in the ice-front advance rate, without altering velocity over the grounding line, but it remains a

useful indicator. On the basis of this, we suggest that the increase in velocity between observations in 1989 and 2000-01 is likely to have occurred in the late 1990s. There are limited oceanic data available to investigate possible changes in oceanic conditions, but a potential mechanism could be changes in ice shelf thickness driven by enhanced basal melting. Indeed this increase coincides with the intense 1997/98 El Nino event which has been linked to abrupt

changes in environmental conditions in the Pacific sector of Antarctica and ice shelf mass loss (Paolo et al., 2018).

### 4.1.2 Drainage of subglacial Lake Cook and short-lived velocity increase

Between the 1$^{st}$ December 2006 and 4$^{th}$ December 2007, the Cook East Ice Shelf was flowing 12% faster at the grounding line than its 2001-2016 average speed (Fig. 5). A similar speed-up

is also evident on the grounded ice upstream and across the entire ice shelf (Fig. 6). This is a greater magnitude of change than expected by interannual variability. In Antarctica, a small number of short-lived accelerations in glacier flow have been observed and linked to subglacial





flood events perturbing basal conditions and leading to enhanced lubrication (e.g. Stearns et al., 2008; Scambos et al., 2011; Siegfried et al., 2016). Between November 2006 and March

2008, we note that a subglacial lake drained ~450 km upstream of Cook East Glacier (Smith et al., 2009; McMillan et al., 2013; Flament et al., 2014), resulting in the discharge of 5.2 ± 1.5 (Flament et al., 2014) or between 4.9 and 6.4 km³ (McMillan et al., 2013) of water, the largest single subglacial drainage event ever recorded. The calculated flow path suggests that the flood could have reached Cook East, but not Cook West (Willis et al., 2016) (Fig. 9). Because the

timing of these two events coincide, we suggest that the acceleration of Cook East Glacier could have been triggered by the drainage of the Cook subglacial lake. The quick response time between the onset of the drainage event and the increase in velocity suggests that at least some of the flood water flowed rapidly through well carved channels, even if some of the floodwater was stored in connecting subglacial lakes (Flament et al., 2014). This adds to the few

observations which link changes in subglacial hydrology to glacier flow dynamics in Antarctica (e.g. Stearns et al., 2008; Scambos et al., 2011; Siegfried et al., 2016). This is important because there are a number of other subglacial lakes which could be routed through Cook East Glacier (Wright et al., 2008). If any changes in subglacial hydraulic conditions occur in the future, the sensitivity of Cook East to perturbations in its basal conditions could be an important

consideration.

### 4.2 Cook West

### 4.2.1 Link between ice shelf retreat and increased velocity

The near-complete loss of the Cook West Ice Shelf (Fig. 7 & 8) is highly unusual in the context East Antarctic outlet glaciers in the past 50 years, where broad trends in their ice-front position

have been linked to climate at decadal timescales, but no other East Antarctic ice shelves have been observed to retreat to their grounding lines and then not re-advanced (Miles et al., 2013; 2016). Our results show that the near-complete loss of the Cook West Ice Shelf between 1973 and 1989 coincided with a likely doubling of Cook West's velocity (Fig. 8). This suggests that the increase in velocity was linked to a reduction in buttressing caused by the loss of the Cook

West Ice Shelf. It would be expected that an increase in velocity of such magnitude would be accompanied by dynamic inland thinning. Indeed, consistent with this notion are satellite altimetry records that, despite covering different time periods between 1992 and 2010, all report an inland thinning signal upstream of Cook West (Davis et al., 2005; Zwally et al., 2005; Shepherd and Wingham, 2007; Pritchard et al., 2009; Flament and Remy, 2012; Schröder et



al., 2018). The thinning signals are modest in comparison to observations in the Amundsen Sea Sector, but we note that these observations were made, in some cases, decades after the loss of the Cook West Ice Shelf. Thinning rates could have been higher in the immediate years following ice shelf retreat, as observed in the Crane Glacier which use to flow into the Larsen B Ice Shelf (e.g. Rott et al., 2018). However, from 2010 onwards inland thinning upstream of

Cook West appears to have slowed down or ceased (McMillan et al., 2014), suggesting that the system might be approaching equilibrium following the loss of the Cook West Ice Shelf.

### 4.2.2 Behaviour of Cook West post ice shelf loss

Since the near-complete loss of the Cook West Ice Shelf, the ice-front has fluctuated by ~3 km, but there have been no signs of a substantial re-advance (Fig. 7 & 8). As a consequence of the

increase in Cook West's velocity following the retreat of its ice shelf, its strain rate near the ice-front will have increased (Benn et al., 2007). This may explain the absence of a re-advance because the increase in strain rate has resulted in an increase in the calving rate. However, because the ice-front position fluctuates by ~3 km it suggests that other external factors may also be important in stabilising the ice front position.

The retreat of the Cook West Ice Shelf resulted in the formation of an embayment, which has been growing in size as the neighbouring Cook East Ice Shelf advanced (Fig. 1). This embayment is typically filled with landfast sea-ice, which may act to stabilize ice tongues (Massom et al., 2010). Conversely, sea-ice break-out events have been linked to major instability and calving events elsewhere in East Antarctica (Miles et al., 2017). Whilst we

observe calving events with sea-ice present, and leading to the build-up of ice mélange at the ice-front, the continuous presence of landfsat sea-ice and mélange appears to be important in enabling ice-front advance (Fig. 10). Between 2009 and 2013 the Cook West ice-front maintained approximately the same position (Fig. 10a), suggesting that repeated calving events prevented ice-front advance. Using the MODIS Worldview viewer, we observe multiple sea-

ice break-out events during this time period. In contrast, between 2014 and 2016 the ice-front advanced ~3 km, during which we observe no break-out events and see that landfast sea-ice and mélange were continuously present at the ice-front (Fig. 10b). This suggests that the backpressure applied by the landfast sea-ice and mélange was enough to limit calving and enable ice-front advance. This behaviour is similar to seasonal ice-front fluctuations of some

outlet glaciers in Greenland, where the seasonal formation of mélange inhibits calving resulting in ice-front advance (e.g. Amundsen et al., 2008; Todd and Christoffersen, 2014). The annual





resolution of our data makes it difficult to determine if these fluctuations in ice-front position have a direct effect on the velocity of Cook West because calving events occur on a sub-annual scale. Future investigation into this process is important because the interaction between ice-front position, landfast sea-ice, mélange and ice dynamics, following the loss of ice shelves is poorly understood in Antarctica, and might be an important process missing in current numerical models simulating future sea level contributions from the ice sheet (e.g. Golledge et al., 2015; DeConto and Pollard, 2016). The recent behaviour of Cook West could be one of the clearest modern-day observations for this process.

### 4.2.3 What caused the retreat of the Cook West Ice Shelf?

The widespread retreat of outlet glaciers in the Antarctic Peninsula (Cook et al., 2016) and the collapse of the Larsen B Ice Shelf (e.g. Scambos et al., 2003) have been linked to an increase in surface air temperatures and warm ocean forcing; while the rapid thinning of ice shelves in the Amundsen Sea Sector and at Totten Glacier have been linked to intrusions of modified Circumpolar Deep Water (mCDW) (e.g. Jenkins et al., 2010; Rintoul et al., 2017). Satellite and modelled estimates of the present day basal melt rate of the remaining Cook East Ice Shelf are low suggesting that, on average, it receives a relatively weak ocean heat source (Depoorter et al., 2013; Rignot et al., 2013; Kusahara et al., 2017). Given the proximity of Cook East to Cook West, it is also likely that Cook West also receives a relatively weak oceanic heat source. We also do not observe any surface melt features in the form of supraglacial lakes or channels during our observations and regional ice core records show no long-term trend in accumulation (Goursaud et al., 2017). Thus, these are not the ocean-climate conditions which would typically be associated with the retreat, thinning or catastrophic loss of ice shelves. Therefore, it is likely that the rapid and near-complete loss of the Cook West Ice Shelf between 1973 and 1989 was driven by ocean-climate conditions that were likely quite different from present-day.

Multiple studies point towards a shift in climate towards greater decadal extremes since the mid-20th century in the wider Cook-Ninnis-Mertz region (Fig. 11). Reconstructions of sea surface conditions over the past 250 year show that since 1960 there has been an increase in glacial meltwater as more intense winds enhance mCDW intrusions onto the continental shelf (Campagne et al., 2015). This deviates from the cyclic behaviour of sea surface conditions driven by the periodic formation of the Mertz polynya in association with the ~70 year calving cycle of the Mertz Tongue (Campagne et al., 2015; Giles, 2017). Reconstructions of ice discharge of the region from marine sediment cores west of Mertz show an increase in ice




discharge ~1980, the magnitude of which might be unprecedented throughout the Holocene
(Crespin et al., 2014).  This was linked to an out of phase calving event of the Ninnis Glacier
(Crespin et al., 2014), but we suggest the increase in discharge of Cook West following the
loss of its ice shelf may have also contributed to this recorded increase in ice discharge.

In addition, a climate coupling exists between the Cook-Ninnis-Mertz region and New
Zealand's glaciers, whereby large-scale atmospheric waves connect the two regions (Crespin
et al., 2014; Mackintosh et al., 2017). The onset of the rapid retreat of mountain glaciers in
New Zealand occurred around the 1940s; this retreat continued at varying rates until the 1990s,
when glaciers advanced in response to regional cooling (Mackintosh et al., 2017). A similar
trend is seen in the Cook-Ninnis-Mertz region; along with Cook West (Fig. 8), Ninnis Glacier
underwent a major retreat in the 1940s (Frezzotti et al., 1998) and there was a switch from
dominant outlet glacier retreat across the wider region in the 1970s and 80s to cooler conditions
and glacier advance from 1990 to 2010 (Miles et al., 2013). A similar change in wind pattern
may also be reflected in temperature reconstructions from 1870 to 2010 in the New Zealand
subantarctic islands, which lie directly between the Cook-Ninnis-Mertz region and New
Zealand, where there is an abrupt switch towards a more variable climate from the 1940s
onwards (Turney et al., 2017). Evidence of such variability is also recorded in wind direction
at the nearest research station, Dumont d'Urville, where there was an abrupt shift in the 1990s
towards more easterly winds (Fig. 12). Taken together, analysis of these studies hints at warmer
regional climate during periods of the mid-20[th] century and a cooler climate from the 1990s
onwards. This is consistent with our interpretation that warmer than present ocean-climate
forcing is likely to have driven the rapid retreat of the Cook West Ice Shelf.

At present there have been no subsurface ocean measurements in the immediate vicinity of
Cook West Glacier. However, the local oceanography west of Cook near the Mertz and Ninnis
Glaciers is one of the most extensively studied in Antarctica (Beaman et al., 2011; Kusahara et
al., 2011; Williams et al., 2011; Tamura et al., 2012; Campagne et al., 2015; Aoki et al., 2017).
Numerical modelling has suggested that a key component of the local oceanography in the
Mertz-Ninnis region is the westward advection of warm mCDW from a depression on the
continental shelf in front of Cook glacier (Kusahara et al., 2017) (Fig. 11). The amount of warm
mCDW advected onto the continental shelf from the bathymetric depression is sensitive to both
interannual variability in atmospheric forcing and large changes in the regional 'icescape' (e.g.
calving of the Mertz Glacier Tongue) (Cougnon et al., 2017; Kusahara et al., 2017). Therefore,
the more variable climate in the mid-20[th] century may have resulted in greater mCDW





intrusions. There have been no observations of the bathymetry in front of Cook Glacier so it is not known if there are any connecting troughs to this depression which could facilitate the delivery of warm mCDW intrusions towards Cook West Glacier. However, given the proximity

of a potential ocean heat source to the Cook West Glacier and the absence of any other obvious drivers, we suggest that periodic mCDW intrusions forced by a more variable climate could have been important in driving the rapid retreat of the Cook West Ice Shelf.

## 5. Conclusion

We have shown that despite little change over the past decade, there have been dynamic changes in the velocity of both the Cook East and West glaciers during periods over the past ~45 years. For Cook East we provide one of the few observations linking a short-lived increase in velocity to a subglacial flood event, in addition to a longer-term velocity increase of 19% between 1989 and 2001. For Cook West we link a doubling of its velocity to the near-complete

loss of its floating ice shelf between 1973 and 1989, which may have been forced by a more variable climate in the mid-20th century. Since the loss of the Cook West Ice Shelf, there have been no signs of a comparable re-advance, but small cycles in ice-front position appear to be linked to sea-ice conditions.

The changes we observe highlight the importance of extending observational records of glacier

change in Antarctica, which are typically confined to satellite altimetry and velocity measurements from the mid-1990s onwards. It is possible that in regions where there is multi-decadal climate variability, this may not be a long enough time-period to assess the sensitivity of outlet glaciers to changes in climate. In the case of Cook West, the changes in velocity we observe in response to the loss of its floating ice shelf are some of the largest recorded in the

satellite era in Antarctica. However, in terms of observations of subsurface ocean temperatures, bathymetry and bed topography, it is one of the least studied. This needs to be addressed in order to fully understand the processes driving changes in the recent past and improve our understanding on how it will respond to future changes in climate. This is important because Cook Glacier drains a large proportion of the Wilkes Subglacial Basin and may have the

potential to make future rapid sea level contributions.



*Data Availability:* Landsat and ARGON imagery along with the aerial photography used in this study are freely available from Earth Explorer. Cosi-Corr is an ENVI plug-in and can be

freely downloaded via its webpage. GEBCO bathymetric data is available to download from https://www.gebco.net/. Meteorological data from Dumont d'Urville is available from the SCAR MET READER. Ice-front position shapefiles and velocity grids are available from the corresponding author.

*Author Contributions:* BM conceived the study, designed and executed the method presented in the research, conducted the analysis, and drafted the original manuscript. All authors discussed the results and contributed towards editing the manuscript.

*Acknowledgments*: This work was partly funded by the Natural Environment Research

Council (grant number: NE/R000824/1) and partly by a Durham University Doctoral Scholarship. Landsat imagery was provided free of charge by the US Geological Survey Earth Resources Observation Science Centre. We thank the DigitalGlobe foundation for the provision of World-View imagery free of charge. We thank E.L.Pope for proving subglacial lake drainage basins. We thank P. Chirstoffersen and J. Lloyd for valuable discussions.






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

Note that Cook East and West drain a large proportion of the Wilkes Subglacial Basin.

600




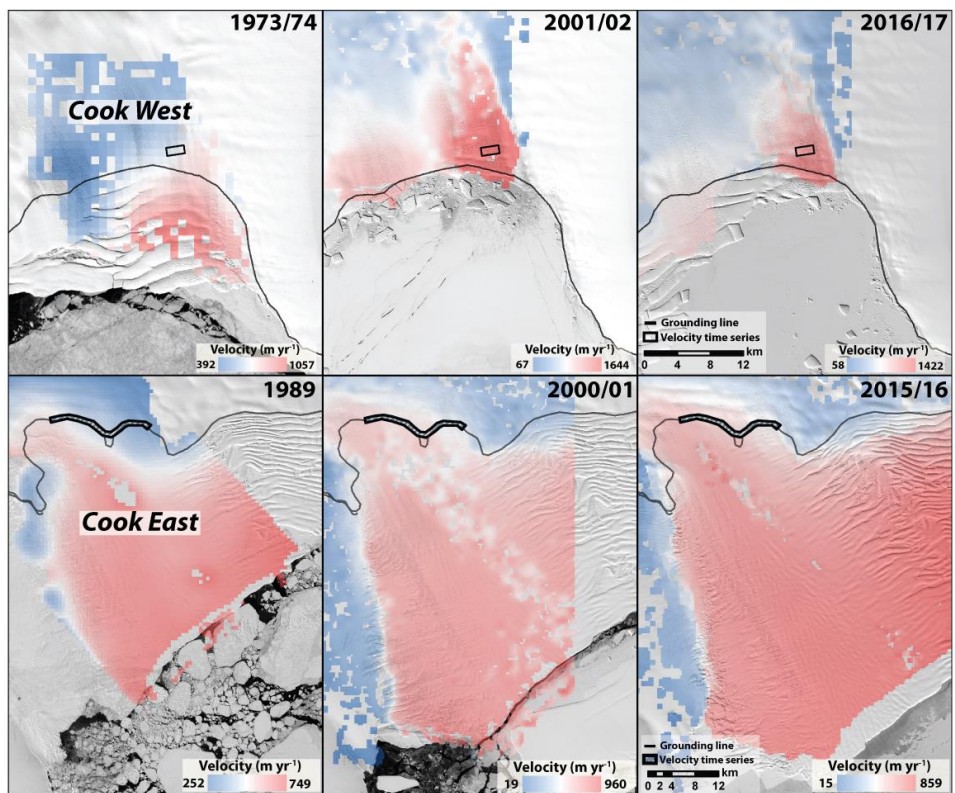

**Figure 2:** Examples of some of the velocity fields calculated in this study for Cook West and
Cook East. Velocity time-series were calculated by taking the average value of pixels within
the respective boxes.

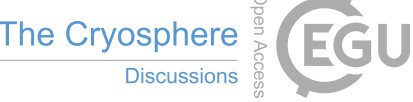



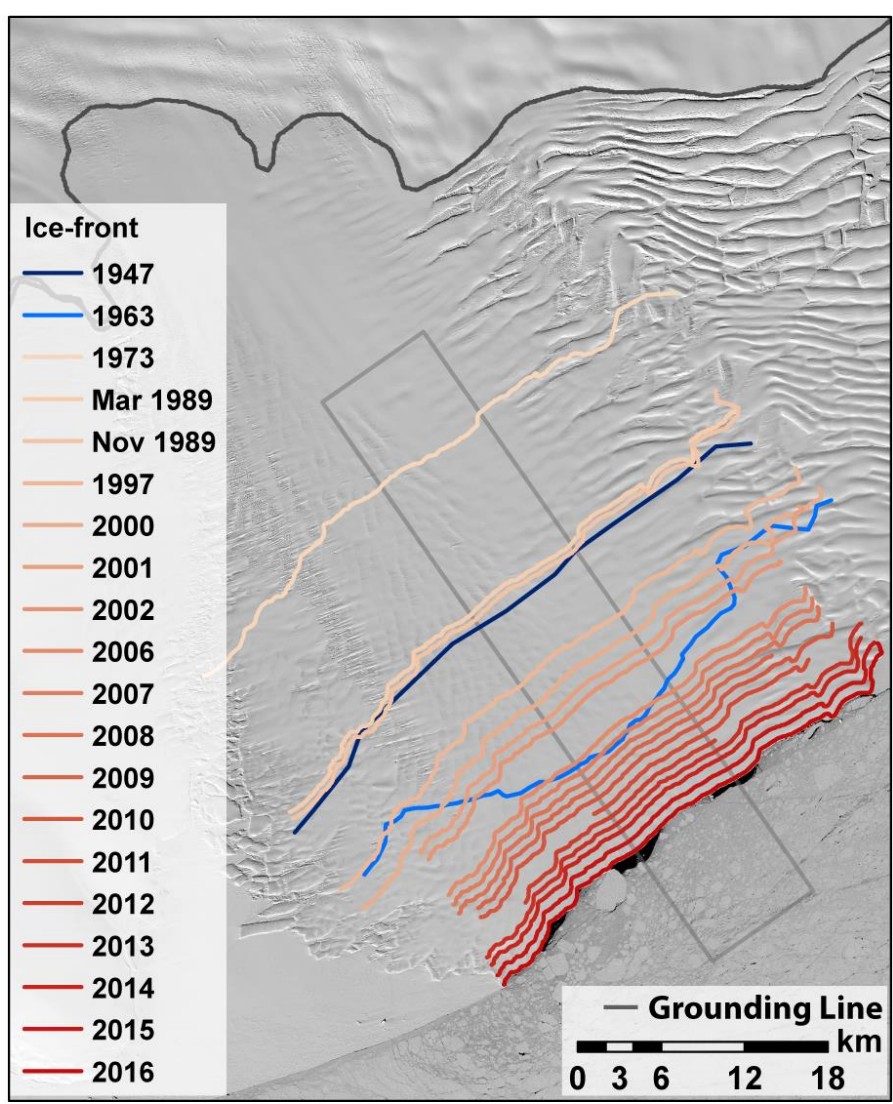

**Figure 3:** Mapped ice-front position of the Cook East Ice Shelf between 1947 and 2016.
Grey box delineates the region where ice-front position change was calculated. Grounding
line is from the MEASURES dataset (Rignot et al., 2011a).



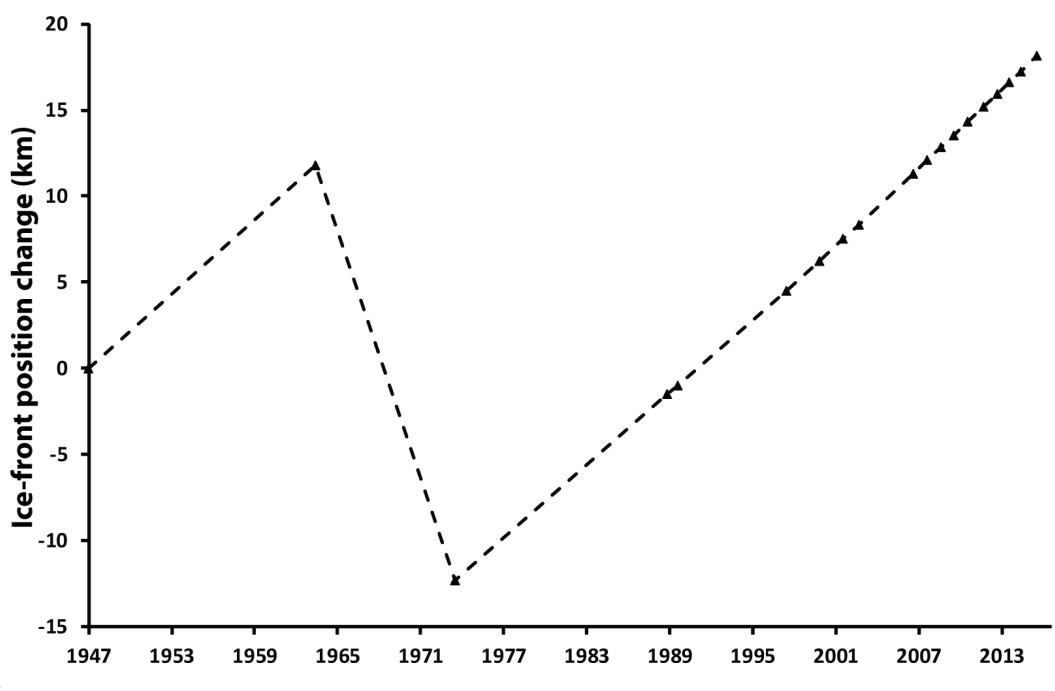


**Figure 4:** Time-series of ice front position change of the Cook East Ice Shelf relative to 1947.






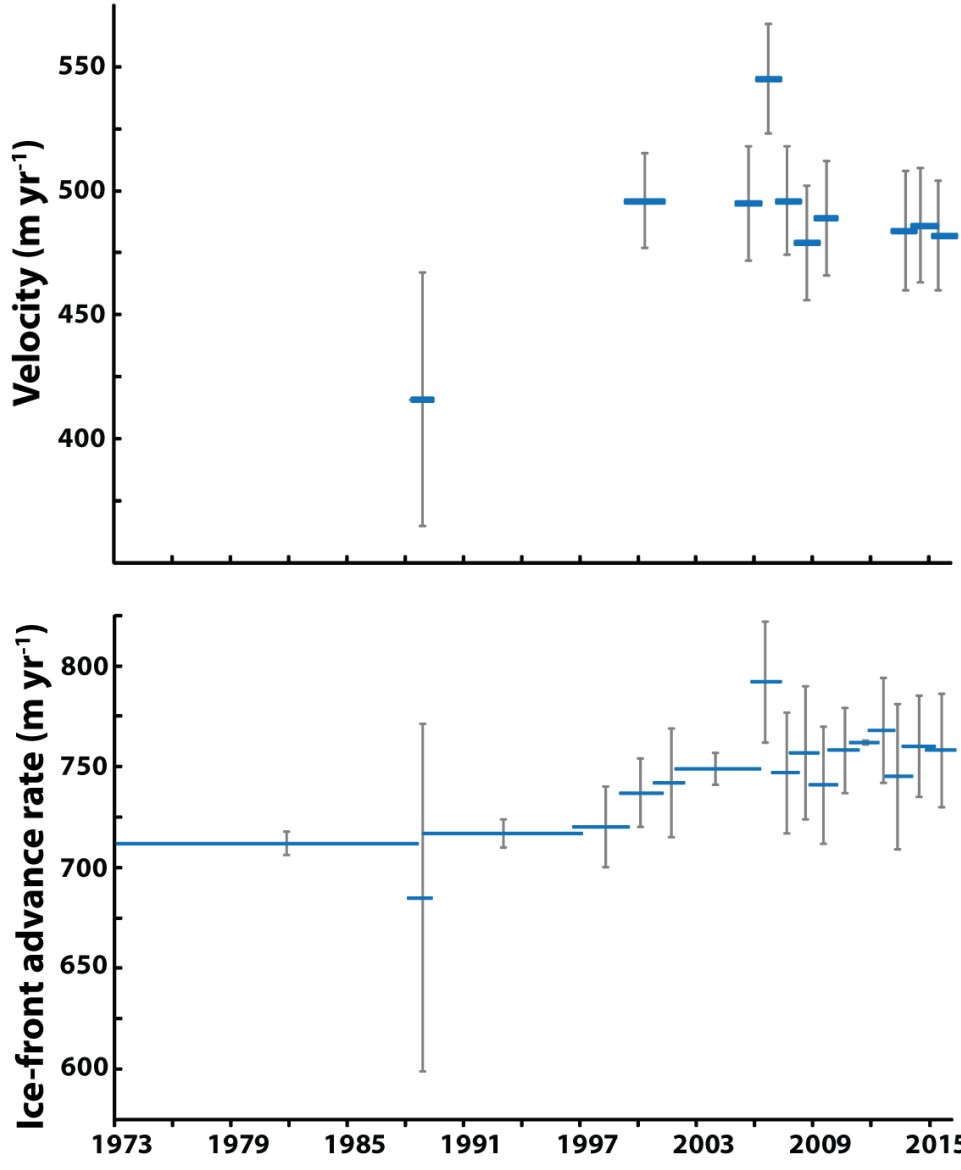

**Figure 5:** Mean velocity derived from feature tracking extracted from the grounding line of Cook East (Top). Ice-front advance rate of the Cook East Ice Shelf (Bottom). Grey bars represent error on both the top and bottom panels.



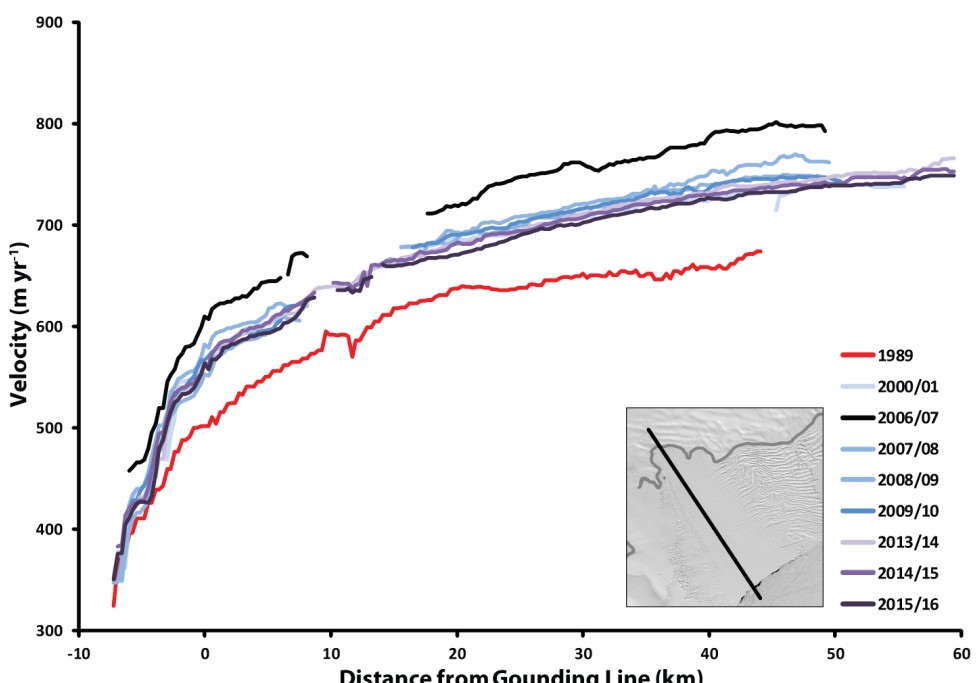


**Figure 6:** Cross-profile of the velocity of the Cook East Ice Shelf. Velocities were extracted along the same series of points shown on the inset.








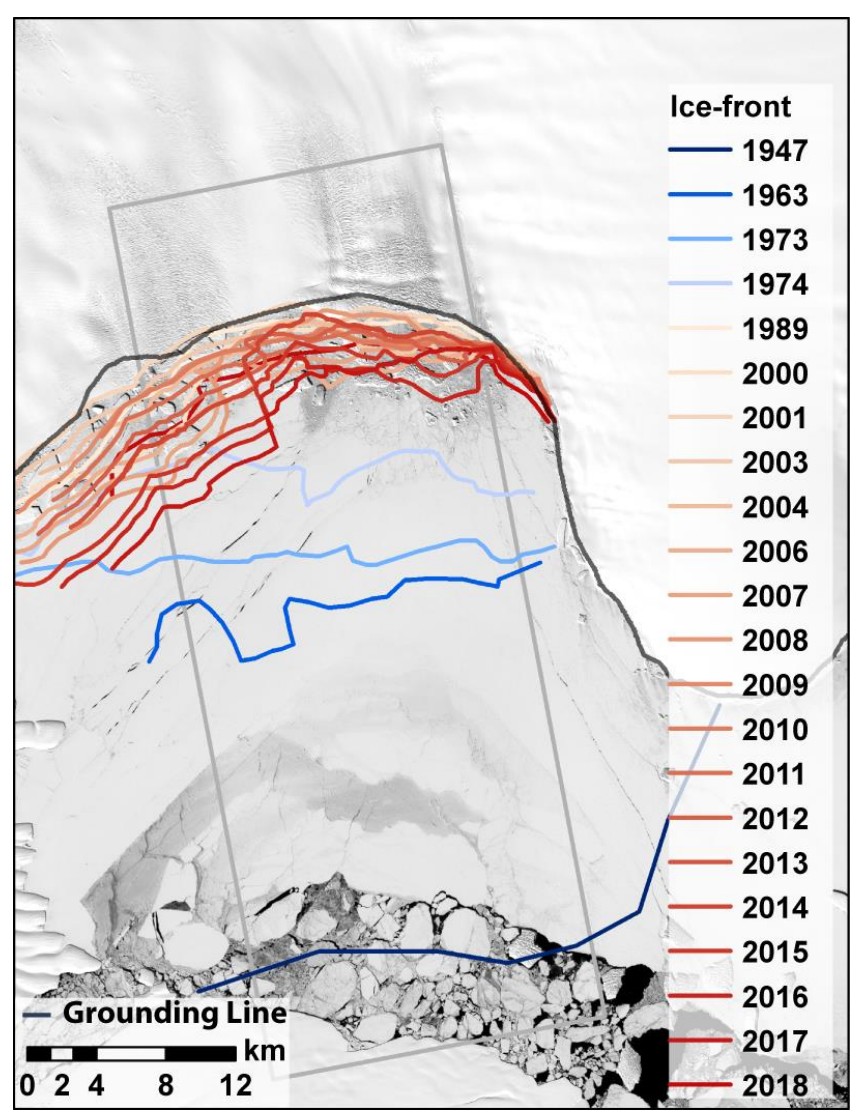

**Figure 7:** Mapped ice-front position of the Cook West between 1947 and 2018. Note the exceptional retreat between 1947 and 1989. Grey box delineates the region where ice-front position change was calculated. Grounding line is from the MEASURES dataset (Rignot et al., 2011a).




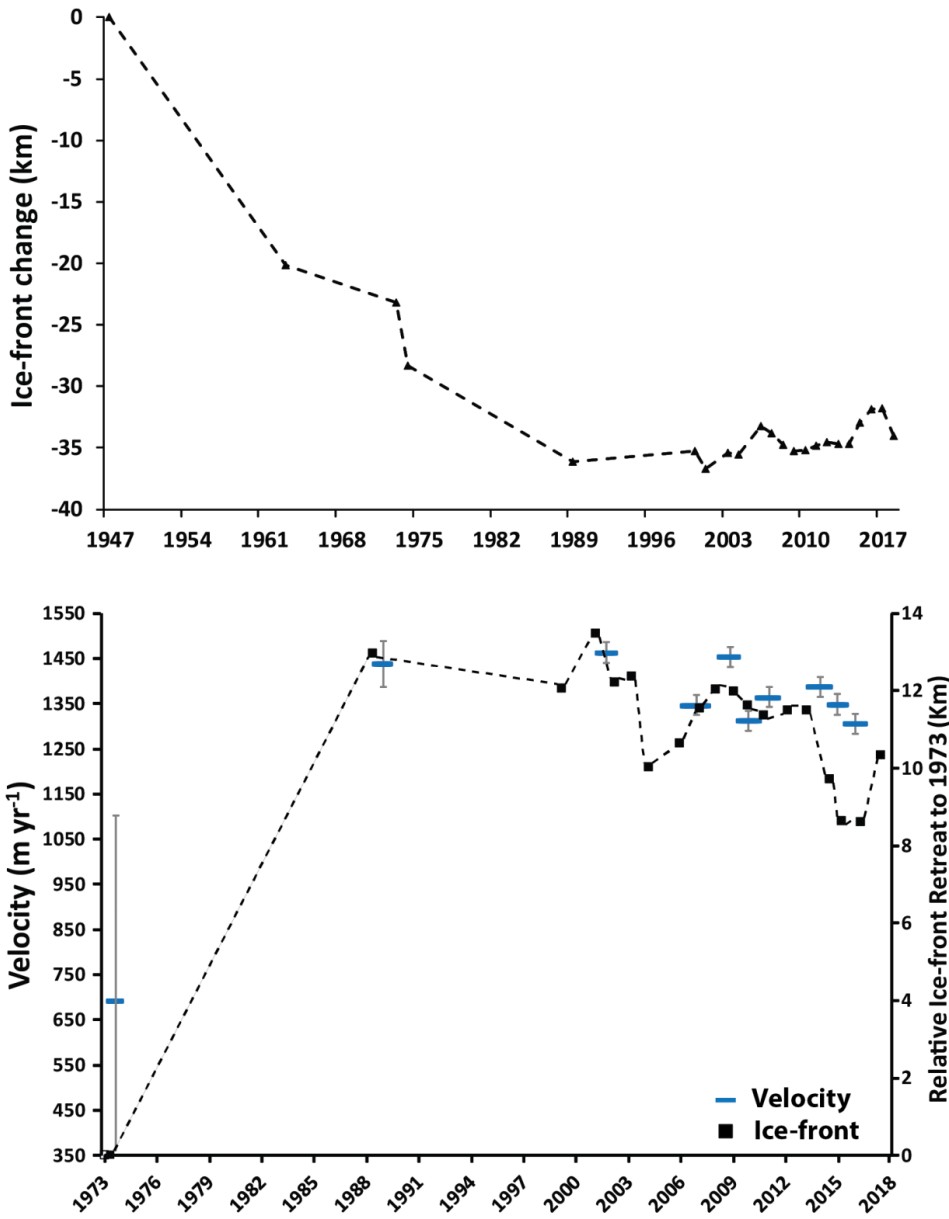

**Figure 8:** Ice-front position time series of Cook West between 1947 and 2018 (Top). Velocity
estimates and ice-front retreat of Cook West Glacier between 1973 and 2017 based on feature-
tracking. The blue bars represent velocity and extend over the period of measurement, grey
bars represent error and the dashed line represents the relative ice-front retreat to 1973. Note
the increase in velocity between 1973-74 and 1989 coincides with the retreat of the Cook West
Ice Shelf.



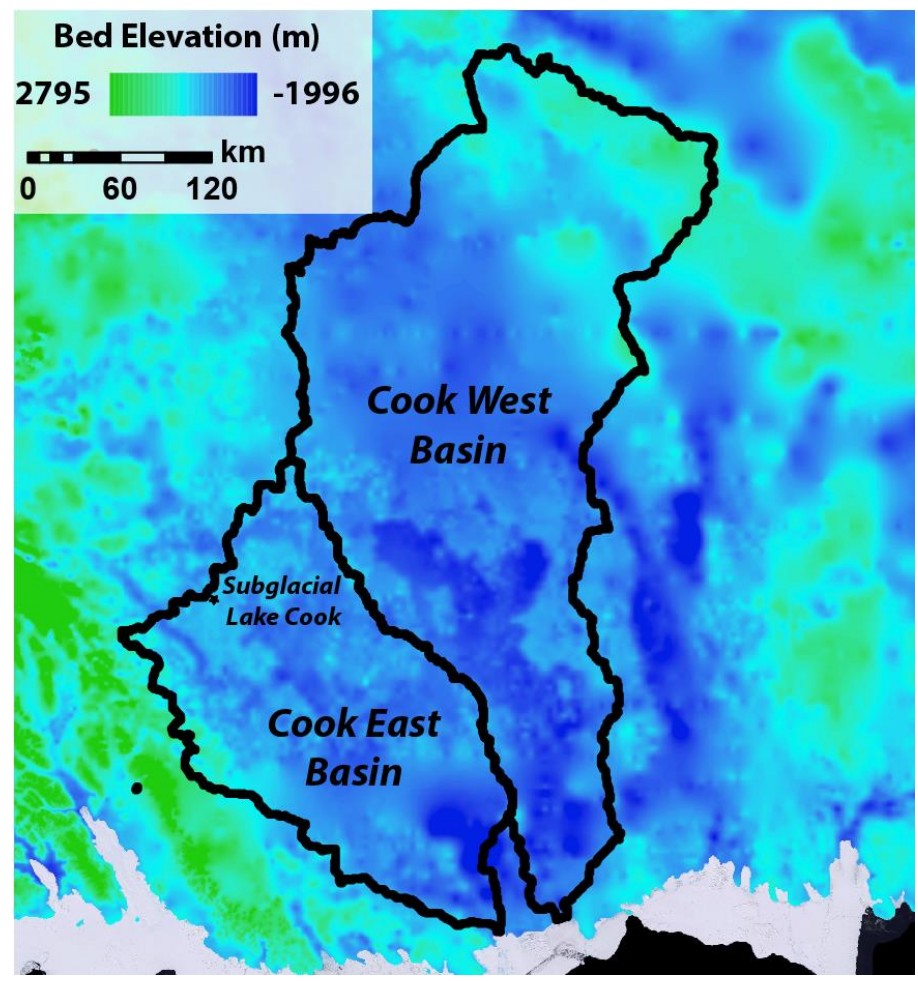


**Figure 9:** Subglacial drainage basins of Cook East and Cook West (Wills et al., 2016) overlain on bed elevation from Bedmap-2 (Fretwell et al., 2013). Subglacial Lake Cook is located within the Cook East basin.






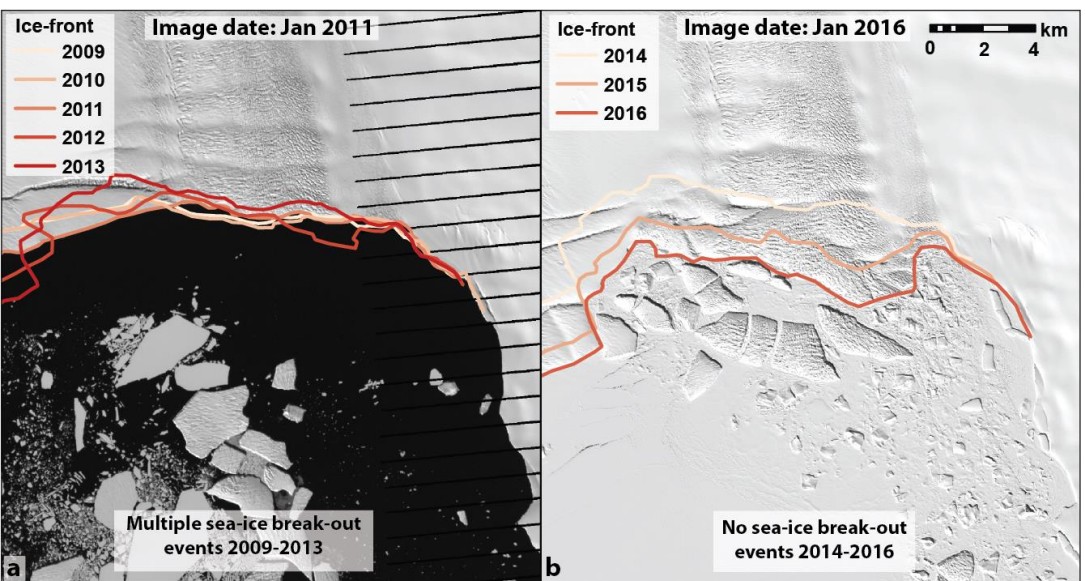


**Figure 10:** Relationship between ice-front position of Cook West and the presence of
landfast sea-ice and mélange at its ice-front. **a)** Mapped ice-front position between 2009 and
2013 during which multiple sea-ice break-out events were observed and there is little change
in ice-front position. **b)** Mapped ice-front position between 2014 and 2016 during which no
sea-ice break-out events were observed, and the ice-front was able to advance. Note the
build-up of ice mélange near the ice-front.





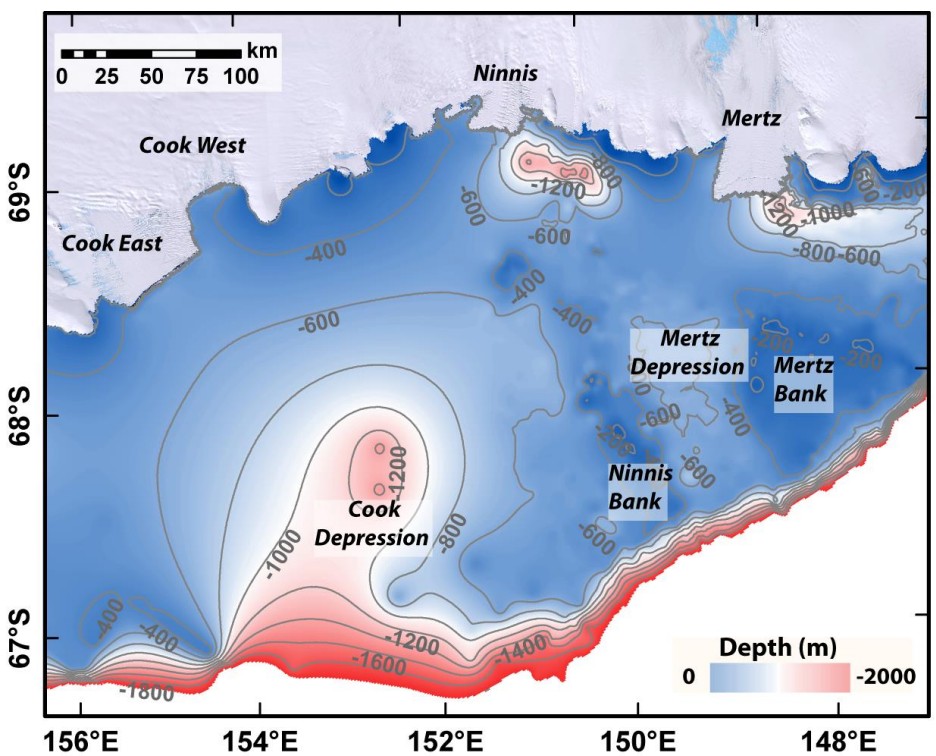


**Figure 11:** General Bathymetric Chart of the Oceans (GEBCO) bathymetry of the Cook-Ninnis-Mertz region overlain on the LIMA mosaic. Note location of the Cook Depression on the continental shelf.






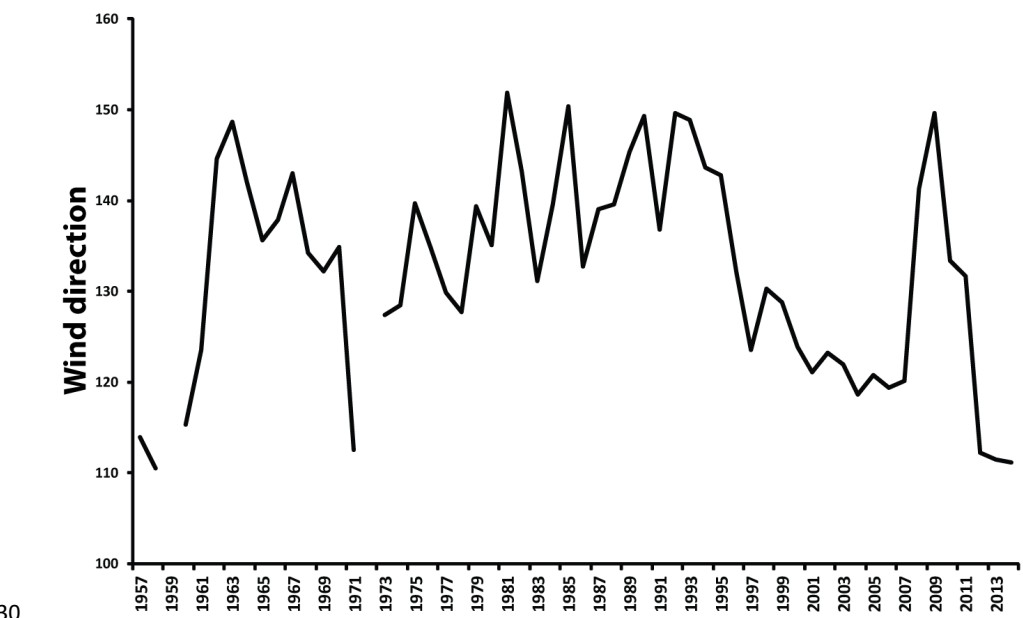

**Figure 12:** Mean annual wind direction from Dumont d'Urville research station 1957-2014.

