# Peer review of "Velocity increases at Cook Glacier, East Antarctica linked to ice shelf loss and a subglacial flood event"

_The Cryosphere, 2018_

## Referee Comment (RC1) · T.ÂăA. Scambos (Referee) · 8 Jul 2018

Review, Miles-Stokes-Jamieson TCryo Cook Glacier This is a good observational paper on the detailed recent history and unique events occurring on the Cook Glacier / Ice Shelf system. The system provides a number of insights into Antarctic glacier systems as additional examples of processes seen elsewhere.

I recommend publishing the paper with minor revisions. Mostly I would like to see some additional information and a few adjustments to the figures. The paper could almost go in as is, but a few extra steps would present the work better and satisfy the curiosity of the reader a bit more.

[Figure]

L22 'subglacial' should be capitalized.

L92 Kääb will need its umlauts.

L140-147 – Note the implications for the current calving of Larsen C. Some have jumped on this retreat into the embayment as an indication of the beginning of an irreversible retreat, and yet Cook East appears to be cycling back. The Larsen C stability question is an important debate these days, and may intensify with an upcoming Rignot paper.

L148 – with your error bars, just say 'approximately 20%'. Your 1989 estimate has an error bar of $\pm12\%$.

L178 change to '...measurement is high (don't need to repeat error here), the pair of measurements still demonstrates a major increase in velocity, which ....'

L187-188 you already said part of this. How about: "The calving of the Cook East Ice Shelf between 1963 and 1973 was unusual in the context of large Antarctic ice shelves where calving events...."

L199-200 Similar note - - just move on: "The increase in velocity between 1989 and 2000-2001 (416 to 496 if you want) coincides with an increase in the ice front advance rate. Notably, most of this increase is concentrated between..."

L253 – '...which use(d) to flow into...' this is colloquial, not for written text. '..which formerly flowed into.'

L281-285 as you know, Landsat 8 acquisitions since 2013-2014 are far better than annual now; the next fast ice break-out will be an interesting study for you (note that GoLIVE data may help as this future event unfolds, https://nsidc.org/data/golive)

L350-353 this section is a good walk through the available climate information and related studies. One thing that would support a sub-ice-shelf melting explanation for the 1970s retreat of Cook West I.S. would be channeling or a change in the character

of the bottom crevasses and rifts. This could be discussed in a bit more detail (is there evidence or no evidence) with the early images in a figure related to Figure 7 (see below).

Figure 1 - I'm feeling a bit too 'zoomed in' here, although it covers the study area well, I think for Figure 1 a slightly larger view would be good, or perhaps a third panel – Antarctica outline, then coast and near-coastal Wilkes Land from Adare to Dumont D'Urville with Bedmap data (in more detail than the current inset) and then the flow speed figure as you have it.

Figure 4 and 5 – consider combining these into one 3-panel figure.

Figure 6 – difficult to tell the difference between 2006/07 and 2015/16 in the graphic, although the text makes it clear that the jump in speed is in 2006/07. A different color scheme would solve this.

Figure 7 – I think you might want to include an extra figure, with the best-contrast versions of the 1947, 1963, 1973 or 1974, and 1989 images – I think the structural details in the ice shelf and the grounded portion of the lower Cook West glacier might provide some insight into the break-up causes – or at least eliminate some.

Figure 8 —lower panel, I would adjust the y-scales to separate these two data sets slightly. The graphic is confusing with such close overlap. It's nice to see the correlation, and the point of the graphic is well taken, but the graph appears at first to be showing some kind of fit or second data set for velocity. The ice front retreat curve is a repeat of the upper panel data for 1973 – present?

Figure 9 – please show the proposed drainage path as determined by Flament et al. 2014 m (their figure 7) – this can be as a grey shaded strip on the bedrock mapping.

Figure 10 please provide the source of the image data (Landsat 7 and Landsat 8?)

Figure 12 interesting plot. An image or line map of Dumont D'Urville as an inset or additional panel would be good to see, with directions (0, 90, 180, 270) marked unobtrusively. The image or line map should include the coast of Cook Glacier as well. Checking the available AWS in the area – it appears that there may be some data from the Russian base Leningradskaya that is closer (a little) than Durmont D'Urville.

---

## Referee Comment (RC2) · Anonymous Referee #2 · 18 Jul 2018

**Summary**

This is an interesting paper, which relies on remote sensing to investigate velocity and ice front position of Cook ice shelves. The text is generally well written and structured and the images are relatively clear. However, I recommend that a few things are addressed/clarified before the manuscript can be recommended for publishing.

**General comments**

-I am surprised to find no mention of grounding line migration. Much emphasis is put on calving front location and velocities, which both play an essential role in ice-shelf stability. However, grounding line position is equally –if not more – important, when it

comes to ice-shelf stability and loss of buttressing. All the more so, given that Cook is a marine basin with a retrograde slope.

Is there a particular reason why only optical data are used to derive velocities and not SAR images? The latter could have helped you to overcome the lack of suitable data.

Have you considered the role of pinning points? You mention ice rises when it comes to co-registering the images but would it be possible that the observed acceleration is somehow linked to a loss of contact with a pinning point (ice rise or ice rumple)?

Have looked at strain rates and their evolution? I think it could provide valuable information about dynamic changes over the period you cover.

I am surprised to see that the paper from (Liu et al., 2015) is not cited and the type of caving occurring at Cook ice shelves (tabular vs disintegrating) not discussed.

How do your velocity fields compare to (Rignot, Mouginot, Scheuchl, 2011)? You only mention this field when you remove outliers but not when you assess your velocity fields.

You talk about an increase in velocity but does this acceleration appear everywhere? Does it vary spatially? You only show changes over one velocity profile in Fig 6.

I feel that that the overall number of figures in the main paper could be reduced, as not all of them are highlighting essential information. For instance, Fig 3 and Fig 7 could be 2 subplots of a same figure. Or Fig 2 and 12 are not really exploited in the main paper so could go in the supplementary.

Generally, I think it would be good to link a bit more the processes occurring in both ice shelves in the discussion. Given their proximity and the similarity in their configuration, it is surprising to find different behaviours. For instance, in section 4.2.3 you conclude that the significant retreat of Cook West's front is probably due to intrusion of mCDW but then how do you explain that Cook East didn't experience the same retreat? (you state yourselves that the ocean source is probably the same for both ice shelves).

**Specific and technical comments**

L27 : "Ice which is grounded well below sea level in the marine basins of Antarctica is potentially vulnerable to marine ice sheet instability." To trigger a marine ice sheet instability, you need two conditions: 1) a bed grounded below sea-level and 2) and retrograde slope (i.e. an inland-sloping bed. It would be good to state this fact more clearly in the text. A bedrock below sea level is not sufficient alone to trigger a marine ice sheet instability. I find that your phrasing here is somehow ambiguous.

L39: "WAIS" is it really useful to introduce this acronym that is used only once?

L44: "substantial" can you give an order of magnitude?

L49: "SLE" acronym not defined

L76, L83 : position not "positon"

L78: "Errors using this method" how is it quantified?

L102: coregistration: aren't the orbital data precise enough to co-register the images? (at least for landsat 8)

L105: "Because these features were relatively common" the features 'are' (it's still the case).

L107: "a grid size of 20 x 20 pixels" what do you mean? Is the spacing 20 pixels or the final grid made of 400 pixels in total? Also, what is the final resolution of your velocity fields?

L108: "Error in surface displacement was estimated at 0.5 pixels" how is it estimated? With stable surfaces?

L117: why this value of 450 m/a?

L116: Fig 2. ). Bracket is missing

L144-147. I am a bit lost here. What makes you say that? Why 2015? It's not very

clear from Fig. 4.

L170-172: "This resulted in an estimated total loss of 1,200 km2 of ice shelf between 1947 and 1989. The large retreat of 5 km between November 1973 and January 1974 (Fig. 7 8)" It is visible in Fig 7 but I cannot see the 5km retreat between 1973-1974 on Fig 8. Why?

L188: How can you be sure that the ice shelf has retreated in the constrained section of the embayment? Have you checked on passive shelf map from Fürst et al (2016)? Have you looked at the strain rates? I think that this claim needs to be substantiated.

L202-203: I don't agree with you when you say that "The rate of ice-front advance is not a direct estimate of velocity because there are processes such as longitudinal stretching which can result in changes in the ice-front advance rate, without altering velocity over the grounding line" While it is true that it is not a direct estimate of velocity because icebergs calve off (as you explain in the methodology). It is not correct to suggest that longitudinal stretching shouldn't be included in velocities. This stretching is the main deformation process that occur on an ice shelf and is also what causes such fast ice flow on ice shelves (and some ice streams). It is however true that, because of longitudinal stretching, velocity of the ice shelf front is different from that of the grounding line. Moreover, how does your feature algorithm work? Does it exclude the ice-shelf front? If not, I would assume that contrasts between the ocean and the ice is a good feature to track and therefore that you would get a very reliable velocity data point at the front (in the absence of calving).

L208: "change in ice shelf thickness". Have you checked if they present Cook ice shelf in the supplementary of (Paolo, Fricker, Padman, 2015)?

L213: "ice shelf was flowing 12In find this phrasing confusing as I understand the sentence as " the velocity at the grounding line was faster than the average of the whole ice shelf", which is not what you mean (I think). It could be a good idea to rephrase this sentence.
L221: 5.2 . . . units are missing here

L223: "the calculated flow path" I get what you mean as flow paths are related to drainage basins but this is not what Fig 9 shows.

L250: "thinning signal is modest" which order of magnitude?

L280-285: Nice to see that!

L314: do you mean "discharge SINCE 1980"?

Figure 1:

-I find it confusing to have Cook East and West on the left and right, respectively, as maps are generally oriented towards the north. Have you considered rotating the map to make the North appear at the top? Or adding an arrow pointing towards the north or something?

-It would be nice to have the grounding line also on the left part of the image. Have you considered another grounding line dataset like (Depoorter et al., 2013) or the updated version of the MEASURE dataset (Rignot, Jacobs, Mouginot, Scheuchl, 2013)?

-I don't find the color scale of the overview very helpful: I am not sure it is colorblind-friendly and, given the boundary of the color scale, it is hard to distinguish which part of the basin is below sea level.

-Have you considered delineating Wilkes Basin? (The overview map might be too small to discern anything though).

Figure 2:

-As the limit of the color scale varies for every map, it is hard to compare them.

-What is shown in background? Landsat images?

Figure 4

-It might be good to specify here as well that the ice-front position is taken in the box delineated in Fig 3

-I am confused here: you claim that ice-front advance accelerates (cf Fig 3) but in Fig 4 all what we see is a straight line, which suggests a constant advance.

Figure 5:

-Using the same color as the lines in Fig 3 (or Fig 6), could help identifying the data points you are referring to.

-Have you thought about marking the different periods you're referring to in the text?

Figure 7

-To improve the readability of the figure I would suggest to delineate the grounding line with a dashed line.

Figure 8

-I find this figure a bit hard to read

-I am not sure than the right part (relative ice-front change) of the bottom panel is adding much information. However, if you decide to keep it, you could consider changing the color of labelling, to match the color of the markers.

-It is confusing to have different x-axis boundaries for the top and bottom panels

Figure 9

-Same comment as for the overview in Fig 1

-The star that locates Lake cook is relatively hard to spot, could you make it appear more clearly?

-I also think it would be interesting to delineate the ice shelves

Figure 10

The lightest lines (2009 on the left and 2014 on the right) are not very visible.

Figure 11

It could be a nice addition to delineate the grounding line here

Figure 12

I think units are missing on the y-axis

**Bibliography**

Depoorter, M. A., Bamber, J. L., Griggs, J. a, Lenaerts, J. T. M., Ligtenberg, S. R. M., van den Broeke, M. R., Moholdt, G. (2013). Calving fluxes and basal melt rates of Antarctic ice shelves. Nature, 502(7469), 89–92. https://doi.org/10.1038/nature12567

Liu, Y., Moore, J. C., Cheng, X., Gladstone, R. M., Bassis, J. N., Liu, H., . . . Hui, F. (2015). Ocean-driven thinning enhances iceberg calving and retreat of Antarctic ice shelves. Proceedings of the National Academy of Sciences of the United States of America, 112(11), 3263–3268. https://doi.org/10.1073/pnas.1415137112

Paolo, F. S., Fricker, H. A., Padman, L. (2015). Volume loss from Antarctic ice shelves is accelerating. Science, 348(6232), 327–331. https://doi.org/10.1126/science.aaa0940 Rignot, E., Jacobs, S., Mouginot, J., Scheuchl, B. (2013). Ice-shelf melting around Antarctica. Science, 341(6143), 266–270. https://doi.org/10.1126/science.1235798

Rignot, E., Mouginot, J., Scheuchl, B. (2011). Ice flow of the Antarctic ice sheet. Science, 333(6048), 1427–1430. https://doi.org/10.1126/science.1208336

---

## Author Comment (AC1) · 21 Aug 2018

**Final author comments – TC-2018-107**

We would like to thank both referees for their constructive reviews of our manuscript.

Below, we have responded to each of the reviewers' comments in blue italics. At the end of our response we included a version of the manuscript with all changes tracked.

**Review 1 – Ted Scambos**

Review, Miles-Stokes-Jamieson TCryo Cook Glacier

This is a good observational paper on the detailed recent history and unique events occurring on the Cook Glacier / Ice Shelf system. The system provides a number of insights into Antarctic glacier systems as additional examples of processes seen elsewhere.

I recommend publishing the paper with minor revisions. Mostly I would like to see some additional information and a few adjustments to the figures. The paper could almost go in as is, but a few extra steps would present the work better and satisfy the curiosity of the reader a bit more.

*We are pleased that Dr. Scambos found our work interesting and thank him for providing helpful and constructive comments on our manuscript.*

L22 'subglacial' should be capitalized.

*Amended.*

L92 Kääb will need its umlauts.

*Amended.*

L140-147 – Note the implications for the current calving of Larsen C. Some have jumped on this retreat into the embayment as an indication of the beginning of an irreversible retreat, and yet Cook East appears to be cycling back. The Larsen C stability question is an important debate these days, and may intensify with an upcoming Rignot paper.

*This is a good suggestion. We now include the following sentence in relation to large ice shelf retreats and ice shelf stability: 'However, since this retreat between 1963 and 1973, the Cook East Ice Shelf has re-advanced and has remained stable, showing that a retreat deep into an ice shelf's embayment does not necessarily result in an irreversible retreat. This observation could be an important consideration in improving our understanding on how recent and future large calving events influence ice shelf stability in Antarctica e.g. Larsen C (Jansen et al., 2015).'*

L148 – with your error bars, just say 'approximately 20%'. Your 1989 estimate has an error bar of ±12%.

*Amended.*

L178 change to '. . .measurement is high (don't need to repeat error here), the pair of measurements still demonstrates a major increase in velocity, which . . ..'

*Amended.*

L187-188 you already said part of this. How about: "The calving of the Cook East Ice Shelf between 1963 and 1973 was unusual in the context of large Antarctic ice shelves where calving events. . .."

*Amended. The section has been re-written with some additional sentences relating to Larsen C (see comment above).*

L199-200 Similar note - - just move on: "The increase in velocity between 1989 and 2000-2001 (416 to 496 if you want) coincides with an increase in the ice front advance rate. Notably, most of this increase is concentrated between. . ."

*Amended.*

L253 – '. . .which use(d) to flow into. . .' this is colloquial, not for written text. '..which formerly flowed into..'

*Amended.*

L281-285 as you know, Landsat 8 acquisitions since 2013-2014 are far better than annual now; the next fast ice break-out will be an interesting study for you (note that GoLIVE data may help as this future event unfolds, https://nsidc.org/data/golive).

*Yes GoLIVE and Sentinel-1a/b will enable an interesting study of the next break-out event.*

L350-353 this section is a good walk through the available climate information and related studies. One thing that would support a sub-ice-shelf melting explanation for the 1970s retreat of Cook West I.S. would be channeling or a change in the character of the bottom crevasses and rifts. This could be discussed in a bit more detail (is there evidence or no evidence) with the early images in a figure related to Figure 7 (see below).

*There does not appear to be any obvious changes in the structure of Cook West between 1947, 1963 and 1973 (revised figure 5). However, the ice shelf does look, at least visually, structurally weak with extensive crevassing. This possibly explains the difference in behaviour between the neighbouring East and West Ice Shelves. We have now added this into the discussion in section 4.3.*

**Figures**

**Figure 1** - I'm feeling a bit too 'zoomed in' here, although it covers the study area well, I think for Figure 1 a slightly larger view would be good, or perhaps a third panel – Antarctica outline, then coast and near-coastal Wilkes Land from Adare to Dumont D'Urville with Bedmap data (in more detail than the current inset) and then the flow speed figure as you have it.

*We have modified Figure 1 to include two panels. A) is Bedmap zoomed in over the Wilkes Subglacial Basin, with an inset of its location in Antarctica. B) is the original Figure 1 with the grounding line updated at the request of reviewer 2.*

**Figure 4 and 5** – consider combining these into one 3-panel figure.

*Figure 4 and 5 are now combined into a 3-panel figure.*

**Figure 6** – difficult to tell the difference between 2006/07 and 2015/16 in the graphic, although the text makes it clear that the jump in speed is in 2006/07. A different color scheme would solve this.

*We have amended the colour scheme so that it is easier to distinguish 2006/07 and 2015/16.*

**Figure 7** – I think you might want to include an extra figure, with the best-contrast versions of the 1947, 1963, 1973 or 1974, and 1989 images – I think the structural details in the ice shelf and the grounded portion of the lower Cook West glacier might provide some insight into the break-up causes – or at least eliminate some.

*We now include an extra panel with the images of Cook West in 1947, 1963, 1973 and 1989. This shows that there is no clear evidence of any significant changes in the structure of Cook West between 1947 and 1973.*

**Figure 8** —lower panel, I would adjust the y-scales to separate these two data sets slightly. The graphic is confusing with such close overlap. It's nice to see the correlation, and the point of the graphic is well taken, but the graph appears at first to be showing some kind of fit or second data set for velocity. The ice front retreat curve is a repeat of the upper panel data for 1973 – present?

*Taking into account the comments from reviewer 2, we have removed the second axis on the lower panel 'relative ice-front position change since 1973' to avoid any confusion.*

**Figure 9** – please show the proposed drainage path as determined by Flament et al. 2014 m (their figure 7) – this can be as a grey shaded strip on the bedrock mapping.

*We have now added the region of most likely flow from Flament et al (2014).*

**Figure 10** please provide the source of the image data (Landsat 7 and Landsat 8?)

*Amended.*

**Figure 12** interesting plot. An image or line map of Dumont D'Urville as an inset or additional panel would be good to see, with directions (0, 90, 180, 270) marked un-obtrusively. The image or line map should include the coast of Cook Glacier as well. Checking the available AWS in the area – it appears that there may be some data from the Russian base Leningradskaya that is closer (a little) than Durmont D'Urville.

*We now include an inset of the coast (Cook – Dumont D'Urville) with wind directions marked. As far as we can tell from AWS project (https://amrc.ssec.wisc.edu/), the AWS at Leningradskaya was only installed January 2008. Hence. Dumont D'Urville is preferred because of the longer time-series.*

**Reviewer 2**

Summary: This is an interesting paper, which relies on remote sensing to investigate velocity and ice front position of Cook ice shelves. The text is generally well written and structured and the images are relatively clear. However, I recommend that a few things are addressed/clarified before the manuscript can be recommended for publishing.

*We are pleased that the reviewer found our work interesting and thank the reviewer for the helpful and constructive comments on our manuscript.*

-I am surprised to find no mention of grounding line migration. Much emphasis is put on calving front location and velocities, which both play an essential role in ice-shelf stability. However, grounding line position is equally –if not more – important, when it comes to ice-shelf stability and loss of buttressing. All the more so, given that Cook is a marine basin with a retrograde slope.

*We agree with the reviewer that grounding line migration is another very important aspect of ice-shelf stability and that this could particularly be true for Cook. One method to quantify grounding line migration would be to map changes in the break in slope as a proxy for the grounding line (e.g. Christie et al., 2016). However, because there are multiple breaks in slope visible as Cook East approaches floatation, this method could be problematic and would result in high uncertainties. Indeed, the difficulties in estimating the break in slope from optical imagery at Cook East are illustrated in the MODIS 2004 and 2009 grounding line products. Here, there is large difference in the grounding line position of Cook East in the order of several kilometres. This difference is too large to be geophysical and more likely represents difficulties in estimating the grounding line of Cook East from optical imagery. An alternative method would be to quantify grounding line migration through differential radar interferometry, but we note that this type of analysis is a large undertaking and can often form the basis of papers alone without additional ice-front/velocity analysis (e.g. Totten – Li et al., 2015). We also note additional differential radar interferometry based grounding line estimates would only be available for recent years and we note that a recent paper has already quantified grounding line migration over this period (Konrad et al., 2018). Thus, we have added some sentences which describe the results of that paper.*

Is there a particular reason why only optical data are used to derive velocities and not SAR images? The latter could have helped you to overcome the lack of suitable data.

*We used optical data simply because it provides the longest time series. We considered using SAR data to overcome data gaps, but on searching for imagery there was very little data for the key gaps in our time series e.g. 2006-07 subglacial flood event, early 2000s and 2011/12. We note that these same image gaps can be seen in the available annual velocity mosaics of Antarctica (see https://nsidc.org/data/NSIDC-0720/versions/1).*

Have you considered the role of pinning points? You mention ice rises when it comes to co-registering the images but would it be possible that the observed acceleration is somehow linked to a loss of contact with a pinning point (ice rise or ice rumple)?

*This is a good suggestion. One of the issues with Cook is the lack of bathymetric observations and we do not observe any obviously grounded icebergs. However, it is a possibility that the retreat of Cook West could be connected to a loss of contact with a pinning point and we now state this in the discussion in section 4.3.*

Have looked at strain rates and their evolution? I think it could provide valuable information about dynamic changes over the period you cover.

*We have not looked at strain rates and feel that this would be beyond the scope of the current manuscript. Moreover, the large error associated with the 1973/74 velocity field would make any strain rate comparison difficult.*

I am surprised to see that the paper from (Liu et al., 2015) is not cited and the type of caving occurring at Cook ice shelves (tabular vs disintegrating) not discussed.

*We now discuss the difference in calving types at the Cook ice shelves with reference to Liu et al. (2015) in section 4.3. Liu et al. (2015) link disintegration-type calving (e.g. Cook West) to enhanced basal melt and tabular calving events (e.g. Cook East) to neutral or positive mass balance regimes. However, given that the Cook East and west ice shelves are in such close proximity and therefore are likely to receive similar climate forcing; we suggest the difference calving type could be related to the structure of the ice shelves which might be related to bed topography as the ice shelves approach floatation.*

How do your velocity fields compare to (Rignot, Mouginot, Scheuchl, 2011)? You only mention this field when you remove outliers but not when you assess your velocity fields.

*We now include a comparison cross-profile to the MEASURES dataset in the revised Figure 4. This shows that our data compares well.*

You talk about an increase in velocity but does this acceleration appear everywhere? Does it vary spatially? You only show changes over one velocity profile in Fig 6.

*In addition to the velocity profile we already show changes in velocity over a section of the grounding line and across the ice-front. All three of these time-series over different parts of the ice shelf show a consistent acceleration.*

I feel that that the overall number of figures in the main paper could be reduced, as not all of them are highlighting essential information. For instance, Fig 3 and Fig 7 could be 2 subplots of a same figure. Or Fig 2 and 12 are not really exploited in the main paper so could go in the supplementary.

*We reduce the number of figures by moving Figure 2 to the supplement and combining Figures 4 & 5 to a single three panel figure. We feel all the other remaining figures are highlighting essential information.*

Generally, I think it would be good to link a bit more the processes occurring in both ice shelves in the discussion. Given their proximity and the similarity in their configuration, it is surprising to find different behaviours. For instance, in section 4.2.3 you conclude that the significant retreat of Cook West's front is probably due to intrusion of mCDW but then how do you explain that Cook East didn't experience the same retreat? (you state yourselves that the ocean source is probably the same for both ice shelves).

*This is an interesting point which we now address in the main text through an additional paragraph in section 4.3. Essentially, we argue that because both ice shelves are so close to each other, they must receive a similar ocean forcing. Thus, we suggest that their difference in behaviour could be driven by different conditions at the bed as the ice shelves approach floatation, leading to different ice shelf structures and calving behaviour. We note that it is difficult to interpret the tabular calving behaviour of Cook East because it is not known what constitutes its typical or natural calving cycle because the length of its calving cycle is longer than our observational record. In some cases large tabular calving events have been attributed natural advance and retreat cycles connected to internal stresses e.g. Amery Ice Shelf (Fricker et al., 2002). However, it is not fully understood how tabular*

*calving events respond to climate variability. We argue that there are hints that Cook East's current calving cycle is different to its last e.g. its current ice-front positon is ~6 km further advanced than its last calving event and its present day morphology indicates that a calving event is at least a few years away.*

**Specific and technical comments**

L27 : "Ice which is grounded well below sea level in the marine basins of Antarctica is potentially vulnerable to marine ice sheet instability." To trigger a marine ice sheet instability, you need two conditions: 1) a bed grounded below sea-level and 2) and retrograde slope (i.e. an inland-sloping bed. It would be good to state this fact more clearly in the text. A bedrock below sea level is not sufficient alone to trigger a marine ice sheet instability. I find that your phrasing here is somehow ambiguous.

*We have amended the text to clarify that marine ice sheet instability requires a bed grounded below sea level and a retrograde slope.*

L39: "WAIS" is it really useful to introduce this acronym that is used only once?

*We have removed the acronym.*

L44: "substantial" can you give an order of magnitude?

*On the order of ~3 m. This has been added to the text.*

L49: "SLE" acronym not defined

*Acronym defined.*

L76, L83 : position not "positon"

*Amended.*

L78: "Errors using this method" how is it quantified?

*Error from co-registration was quantified by digitizing the difference between stable features in image pairs (1 pixel). The error associated with mapping ice-front has been widely attributed to be 0.5 pixels (e.g. Miles et al., 2013; 2016). This gives a total estimated error of 1.5 pixels. We now detail this in the text.*

L102: coregistration: aren't the orbital data precise enough to co-register the images? (at least for landsat 8)

*Yes, for Landsat 8 the orbital data is precise enough to co-register the images without manual co-registration. For all other image pairs ASTER, Landsat 1,4 and 7 manual co-registration was required. We have now clarified this in the text.*

L105: "Because these features were relatively common" the features 'are' (it's still the case).

*Amended.*

L107: "a grid size of 20 x 20 pixels" what do you mean? Is the spacing 20 pixels or the final grid made of 400 pixels in total? Also, what is the final resolution of your velocity fields?

*The spacing is 20 x 20 pixels. This means that the final resolution of the velocity fields is 20 x the image resolution e.g. Landsat 8 300 m, Landsat 4 600 m etc. We have clarified this in the text.*

L108: "Error in surface displacement was estimated at 0.5 pixels" how is it estimated? With stable surfaces?

*It was estimated from manually tracking large surface features e.g. crevasses, we have amended to text accordingly.*

L117: why this value of 450 m/a?

*This value is chosen because it the estimated error of the Landsat – 1 image pair velocity field.*

L116: Fig 2. ). Bracket is missing

*Amended.*

L144-147. I am a bit lost here. What makes you say that? Why 2015? It's not very clear from Fig. 4.

*We know that Cook East calved at some point between 1963 and 1973, but do not know the exact date. From extrapolating the rate of advance, its maximum possible extent would have been similar to its 2015 position, meaning its current extent must be further advanced than the last point it calved. To simply this we have amended the text to simply say 'Through extrapolating the rate of advance between 1947 and 1963 to establish Cook East's maximum possible extent, it is clear that its present-day ice-front is further advanced than the point at which it last underwent a major calving event/retreat.'*

L170-172: "This resulted in an estimated total loss of 1,200 km2 of ice shelf between 1947 and 1989. The large retreat of 5 km between November 1973 and January 1974 (Fig. 7 8)" It is visible in Fig 7 but I cannot see the 5km retreat between 1973-1974 on Fig 8. Why?

*The 5 km retreat is visible on Figure 8 (top), but we did not include the 1974 measurement on Figure 8b (bottom). The amended Figure 6 does not included the second y axis 'relative ice-front retreat since 1973), so this is no longer an issue.*

L188: How can you be sure that the ice shelf has retreated in the constrained section of the embayment? Have you checked on passive shelf map from Fürst et al (2016)? Have you looked at the strain rates? I think that this claim needs to be substantiated.

*On the revised Figure 2 we now include the passive ice boundary calculated in Fürst et al (2016). This shows that the ice-front positon of Cook East in 1973 was several kilometres into the constrained section of the embayment.*

L202-203: I don't agree with you when you say that "The rate of ice-front advance is not a direct estimate of velocity because there are processes such as longitudinal stretching which can result in changes in the ice-front advance rate, without altering velocity over the grounding line" While it is true that it is not a direct estimate of velocity because icebergs calve off (as you explain in the methodology). It is not correct to suggest that longitudinal stretching shouldn't be included in velocities. This stretching is the main deformation process that occur on an ice shelf and is also what causes such fast ice flow on ice shelves (and some ice streams). It is however true that, because of longitudinal stretching, velocity of the ice shelf front is different from that of the grounding line. Moreover, how does your feature algorithm work? Does it exclude the ice-shelf front? If not, I would assume that contrasts between the ocean and the ice is a good feature to track and therefore that you would get a very reliable velocity data point at the front (in the absence of calving).

*To avoid any unnecessary confusion we now remove the section 'The rate of ice-front advance is not a direct estimate of velocity because there are processes such as longitudinal stretching which can result in changes in the ice-front advance rate, without altering velocity over the grounding line'*

*The feature tracking algorithm does exclude the ice-shelf front. The ice-front advance rate was calculated by manually digitizing the ice-front and is described in section 2.3. We agree that in the absence of calving it does provide a very reliable estimate of velocity.*

L208: "change in ice shelf thickness". Have you checked if they present Cook ice shelf in the supplementary of (Paolo, Fricker, Padman, 2015)?

*We have obtained the data from Paolo et al (2015), it does show a potential ice shelf thinning episode in the late 1990s (coinciding the increase in velocity of Cook East) and considered using it in the manuscript. However because the associated error is high and it is not clear which part of the ice shelf is actually measured, we decided not to include it.*

L213: "ice shelf was flowing 12In find this phrasing confusing as I understand the sentence as " the velocity at the grounding line was faster than the average of the whole ice shelf", which is not what you mean (I think). It could be a good idea to rephrase this sentence.

*We have now simplified this to 'the ice shelf was flowing 12% faster than its 2001-2016 average speed' to avoid confusion.*

L221: 5.2 . . . units are missing here

*Amended*

L223: "the calculated flow path" I get what you mean as flow paths are related to drainage basins but this is not what Fig 9 shows.

*We have now added the region of most likely flow path from Flament et al (2014) to the revised Figure 7.*

L250: "thinning signal is modest" which order of magnitude?

*The thinning signal is in the region of 50 cm/yr. we have clarified this in the main text.*

L280-285: Nice to see that!

*Thanks!*

L314: do you mean "discharge SINCE 1980"?

*Amended*

**Figures**

**Figure 1:** -I find it confusing to have Cook East and West on the left and right, respectively, as maps are generally oriented towards the north. Have you considered rotating the map to make the North appear at the top? Or adding an arrow pointing towards the north or something?

 -It would be nice to have the grounding line also on the left part of the image. Have you considered another grounding line dataset like (Depoorter et al., 2013) or the updated version of the MEASURE dataset (Rignot, Jacobs, Mouginot, Scheuchl, 2013)?

-I don't find the color scale of the overview very helpful: I am not sure it is colorblindfriendly and, given the boundary of the color scale, it is hard to distinguish which part of the basin is below sea level.

 -Have you considered delineating Wilkes Basin? (The overview map might be too small to discern anything though).

*These are all helpful suggestions and based on these and the comments from reviewer 1 we have made amendments on Figure 1. The Figure now has two panels a) is bedmap zoomed into the Wilkes Subglacial Basin with a nicer colour scheme. b) is the original figure with the Depoorter et al. (2013) grounding line and we have also inserted a north arrow to avoid confusion.*

**Figure 2:** -As the limit of the color scale varies for every map, it is hard to compare them. -What is shown in background? Landsat images?

*We have now amended the colour scale so it is consistent for each set of images and clarify that the background images are the corresponding Landsat images. We have now moved this figure to the supplement.*

**Figure 4** -It might be good to specify here as well that the ice-front position is taken in the box delineated in Fig 3

-I am confused here: you claim that ice-front advance accelerates (cf Fig 3) but in Fig 4 all what we see is a straight line, which suggests a constant advance.

*The purpose of Figure 4 is to show the long term calving cycle of Cook East which is an important observation, this requires a y-axis scale range in the order of 35 km. The magnitude of the increase in ice-front advance shown in Figure 5 (Now Figure 3) is in the order 50m/yr. This equates to a ~12% increase in ice-front advance, which is significant, but also would not be clear on a y-axis scale of 35 km. For example, if the ice-front is advancing an additional 50m/yr, over 20 years it would mean an additional 1 km advance than it otherwise would have if its advance rate had remained constant. On a y-axis scale of 35 km an additional advance of ~1 km is relatively small, hence why the increase in ice-front advance is not clear. Essentially, it is a question of scale. We now specify that the ice-front position is taken from the box delimited in Figure 3 (now Figure 2).*

**Figure 5:** -Using the same color as the lines in Fig 3 (or Fig 6), could help identifying the data points you are referring to.

 -Have you thought about marking the different periods you're referring to in the text?

*We do not use the same colour as the lines in Fig. 3. We feel these figures have different purposes and not want to confuse them. Fig 3 (Now Fig.2) shows the longer term evolution of Cook East and because of the scale of the figure it is very difficult to see the acceleration in its ice-front advance rate shown in Figure 5 (Now 3c). We do however note the accelerations referred to in the text in the figure caption.*

**Figure 7** -To improve the readability of the figure I would suggest to delineate the grounding line with a dashed line.

*The grounding line is now a dashed line.*

**Figure 8** -I find this figure a bit hard to read -I am not sure than the right part (relative ice-front change) of the bottom panel is adding much information. However, if you decide to keep it, you

could consider changing the color of labelling, to match the color of the markers. -It is confusing to have different x-axis boundaries for the top and bottom panels

*To simplify this we have now removed the 'relative ice-front change' additional axis from the bottom panel.*

Figure 9 -Same comment as for the overview in Fig 1 -The star that locates Lake cook is relatively hard to spot, could you make it appear more clearly? -I also think it would be interesting to delineate the ice shelves

*We have changed the colour scheme and made the location of Lake Cook clearer. We have also added the most likely flow route of Lake Cook from fig. 7 (Flament et al., 2014) and delineated the ice shelves.*

**Figure 10** The lightest lines (2009 on the left and 2014 on the right) are not very visible.

The colour scheme has been amended to a bright and more visible yellow.

**Figure 11** It could be a nice addition to delineate the grounding line here.

The grounding line has been added.

**Figure 12** I think units are missing on the y-axis

Amended

[revised manuscript text omitted]